# A Comprehensive Exploration of Therapeutic Strategies in Inflammatory Bowel Diseases: Insights from Human and Animal Studies

**DOI:** 10.3390/biomedicines12040735

**Published:** 2024-03-26

**Authors:** Inês Esteves Dias, Isabel Ribeiro Dias, Teresa Franchi-Mendes, Carlos Antunes Viegas, Pedro Pires Carvalho

**Affiliations:** 1CITAB—Centre for the Research and Technology of Agro-Environmental and Biological Sciences, University of Trás-os-Montes e Alto Douro (UTAD), Quinta de Prados, 5000-801 Vila Real, Portugal; ines.e.dias@gmail.com (I.E.D.); idias@utad.pt (I.R.D.); 2Inov4Agro—Institute for Innovation, Capacity Building and Sustainability of Agri-Food Production, Quinta de Prados, 5000-801 Vila Real, Portugal; 3Department of Veterinary Sciences, School of Agricultural and Veterinary Sciences (ECAV), University of Trás-os-Montes e Alto Douro (UTAD), Quinta de Prados, 5000-801 Vila Real, Portugal; 4CECAV—Centre for Animal Sciences and Veterinary Studies, University of Trás-os-Montes e Alto Douro (UTAD), Quinta de Prados, 5000-801 Vila Real, Portugal; 5AL4AnimalS—Associate Laboratory for Animal and Veterinary Sciences, Quinta de Prados, 5000-801 Vila Real, Portugal; 6Department of Bioengineering and IBB—Institute for Bioengineering and Biosciences at Instituto Superior Técnico, University of Lisbon, Av. Rovisco Pais, 1049-001 Lisboa, Portugal; maria.franchi.mendes@tecnico.ulisboa.pt; 7Associate Laboratory i4HB—Institute for Health and Bioeconomy at Instituto Superior Técnico, University of Lisbon, Av. Rovisco Pais, 1049-001 Lisboa, Portugal; 8CIVG—Vasco da Gama Research Center, University School Vasco da Gama (EUVG), Campus Universitário, Av. José R. Sousa Fernandes, Lordemão, 3020-210 Coimbra, Portugal; pedro@vetherapy.co; 9Vetherapy—Research and Development in Biotechnology, 3020-210 Coimbra, Portugal

**Keywords:** cat, dog, human, inflammatory bowel disease, mesenchymal stromal cells, stem cells

## Abstract

Inflammatory bowel disease (IBD) is a collective term for a group of chronic inflammatory enteropathies which are characterized by intestinal inflammation and persistent or frequent gastrointestinal signs. This disease affects more than 3.5 million humans worldwide and presents some similarities between animal species, in particular, dogs and cats. Although the underlying mechanism that triggers the disease is not yet well understood, the evidence suggests a multifactorial etiology implicating genetic causes, environmental factors, microbiota imbalance, and mucosa immune defects, both in humans and in dogs and cats. Conventional immunomodulatory drug therapies, such as glucocorticoids or immunosuppressants, are related with numerous adverse effects that limit its long-term use, creating the need to develop new therapeutic strategies. Mesenchymal stromal cells (MSCs) emerge as a promising alternative that attenuates intestinal inflammation by modulating inflammatory cytokines in inflamed tissues, and also due to their pro-angiogenic, anti-apoptotic, anti-fibrotic, regenerative, anti-tumor, and anti-microbial potential. However, this therapeutic approach may have important limitations regarding the lack of studies, namely in veterinary medicine, lack of standardized protocols, and high economic cost. This review summarizes the main differences and similarities between human, canine, and feline IBD, as well as the potential treatment and future prospects of MSCs.

## 1. Introduction

Inflammatory bowel disease (IBD) is a collective term for a group of chronic inflammatory enteropathies, which are characterized by intestinal inflammation and persistent or frequent gastrointestinal signs for more than three weeks [1,2,3,4]. The etiology of IBD is not fully understood, often being called idiopathic [5,6]. This disease has some differences in classification in humans and companion animals. In humans, IBD is divided in two main disorders, namely ulcerative colitis and Crohn’s disease; while IBD is mostly a disease of young people, it can develop at any age, with roughly 25% of IBD patients presenting before the age of 20 [7]. Clinical signs and symptoms include vomiting, diarrhea, abdominal pain, decreased appetite, and weight loss [1,8]. Currently, there is no curative treatment for IBD, but attenuation of clinical signs is possible with medical treatment if the signs are moderate to mild; however, in more severe cases, such as perianal fistulas, surgical treatment may be required [9,10]. If the disease is not well managed, this increases the risk of developing intestinal cancer, such as colitis-associated-cancer caused by external oncogenic factors [1]. 

Despite the fact that the etiopathogenesis of the disease has not yet been fully clarified, observations in humans and animals propose a combination of genetic causes, environmental factors, microbiota imbalance, and mucosa immune defects [3,4,11,12]. When a person or animal has a genetic predisposition and a combination of these factors, IBD can arise. The disease has some aspects in common between species, but clearly has some differences in classification, treatment, and outcome [2]. In veterinary medicine, the term chronic inflammatory enteropathy (CIE) is preferred over IBD because it acknowledges the differences in clinical signs, treatment requirements, and the need for surgery between dogs and cats compared to humans [2]. Some dog breeds (e.g., Weimaraner, Rottweiler, German Shepherd Dog, Border collie, Boxer, Yorkshire terrier, and French Bulldog) [2,13,14] and cat breeds (e.g., Siamese or Asian) [15] appear to be more predisposed to developing the disease. 

CIE in veterinary medicine is often classified by treatment response as follows: antibiotic-responsive enteropathy (ARE), food-responsive enteropathy (FRE), immunosuppressant-responsive enteropathy (IRE), and as non-responsive enteropathy (NRE) [16,17]. Although the classification remains antibiotic-responsive, there is lack of consensus among authors regarding its use [17]. Protein-losing enteropathy (PLE) occurs when there is protein loss throughout the intestinal wall, resulting in hypoalbuminemia [17]. The term IBD in companion animals is applied to cases unresponsive to food trials, antimicrobial treatments and, in canine colitis, that do not respond to therapies with sulfasalazine or its derivatives, therefore requiring immunosuppressive therapy [5,6].

Due to the fact that the majority of dogs and cats with CIE do not require immunosuppressive therapy, the term CIE is often used instead to refer patients with chronic gastrointestinal signs [2].

CIE can be either classified according to the affected anatomical area (stomach, small intestine, or large intestine) and according to the predominant type of cellular infiltrate present in the intestinal lamina propria (eosinophylic, neutrophylic, granulomatous, and the most common in dogs—lymphocytic plasmacytic) [18,19,20]. Figure 1 resumes the classification of CIE in dogs and cats according to treatment response.

Given the above, the goal of this study is to present a critical perspective, highlighting the key differences and similarities between human, canine, and feline IBD, as well as conventional therapy and innovative research, notably using MSCs and, in particular, adipose-derived mesenchymal stromal cells (ASCs). A scientific literature search was conducted using the primary electronic databases for scientific publication dissemination—PubMed, Scopus, and Web of Science—to compile and discuss the clinical trials using mesenchymal stem/stromal cells therapy for IBD treatment in human and companion animals. For this purpose, from all the articles published within the period between January 2000 to January 2024, referring to the association of the following keywords—dog, cat, human, mesenchymal stem/stromal cells and IBD—in their article title, abstract or keywords, data were extracted, analyzed, and discussed. All other types of articles were excluded, namely retrospective analyzes, preclinical studies in canine or feline models, studies with local infusion of MSCs instead IV infusion, treatment of perianal fistulas and studies published prior to 2000. Finally, a conclusion and future perspectives in regenerative therapies for IBD treatment are presented. Figure 2 illustrates the flowchart of study selection included in this article. 

## 2. Inflammatory Bowel Disease

### 2.1. Etiopathogenesis

According to researchers, the etiology of the disease is likely triggered by a breakdown of immunologic tolerance to luminal antigens, such as commensal microorganisms and food components [5,21,22]. A microbiota imbalance frequently results from the immune system’s inability to discriminate between commensal bacteria and pathogenic bacteria [23,24]. The intestinal mucosal immune barrier comprises gut-associated lymphoid tissue (GALT), secretory antibodies, and mesenteric lymph nodes that respond to toxins, antigens, and potentially dangerous pathogens [25]. Despite exposure to numerous intraluminal microbial and dietary allergens and the presence of a high number of lymphoid cells, the intestine exhibits low physiological inflammation under normal circumstances. This is a result of the interaction between the epithelial mucosal barrier, innate immune system, and adaptive immune system [26]. Figure 3 resumes the principal causes, risk factors, injury sites, and clinical signs of human, canine, and feline IBD.

In humans and animals IBD, there is more permeability of the intestinal barrier due to improper tight junctions and adherent junction control. Immune cells may become more exposed to luminal antigens as a result of decreased barrier function [20,27]. Intestinal Epithelial Cells (IECs) play an important role in regulating the immune response by expressing pattern-recognition receptors (PRRs) for microorganisms, such as Toll-like receptors (TLRs), and by producing a variety of humoral factors, including cytokines [26]. The interaction of IECs, neutrophils, macrophages, dendritic cells (DCs), and eosinophils as well as their secreted products underlies the non-specific innate immune response, which promotes the differentiation of pathogen-associated molecular patterns [28]. Depending on PRRs, one can determine whether the antigens are tolerated or are met with a reaction [20,29]. Neutrophils are one of the first lines of cellular defense; they can remove infections by boosting inflammation through phagocytosis, producing reactive oxygen species (ROS) and by releasing intracellular chemicals known as neutrophil extracellular traps (NETs), promoting mucosal healing and inflammation reduction. Nonetheless, NETs also trigger epithelial cell death, and ROS also damage lipids and proteins in the mucosa barrier, altering their function [25].

The innate immune response culminates in the activation of dendritic cells, which function as a bridge between innate and adaptive immunity [30]. Lymphocytes are the principal effector cells in the adaptive immune response; they are activated in the gut to destroy harmful antigens [28,31]. Antigens are presented by dendritic cells to particular lymphocytes, such as naïve CD4+ T helper cells in secondary lymphoid organs (e.g., the mesenteric lymph node) [20,32]. Depending on the profile of cytokines releases, T helper cells can differentiate into Th1, Th2, and Th17 cells and regulatory T cells (Tregs), which produce multiple cytokines with different actions (Figure 4) [28,32]. After the antigens have been removed, lymphocytes are normally down-regulated to maintain intestinal homeostasis [26]. In IBD, there is a continuous stimulation of these lymphocytes, resulting in chronic inflammation [8], and this disparity leads to the activation of macrophages and B cells, as well as the recruitment of circulating leukocytes into the gut [33]. Some authors could not demonstrate a clear Th1 or Th2 cytokine expression in dogs and cats with CIE [4,15,28,34], and there is no indication of a Th17 signature [35]. This finding in canine and feline CIE differs with human IBD, where patients with Crohn’s disease had relevant Th1/Th17 (cell-mediated) cytokine polarization, and in ulcerative colitis, a significant Th2 (humoral) cytokine profile [4,36]. 

Genetic abnormalities in innate immune system have been linked to IBD susceptibility in both dogs and humans. Mutation in NOD2/CARD15 in humans, and TLR4 and TL5 in dogs, along with the presence of enteric microbiota, may result in increased proinflammatory cytokine production and decreased bacterial clearance, increasing chronic intestinal inflammation. In felines, the IBD pathophysiology has not been fully determined [15,37,38].

### 2.2. Conventional Therapies

Medical therapy is essential in the treatment of human IBD and aims to reduce inflammation. On the other hand, the majority of dogs and cats with CIE do not require any therapy other than dietary changes (e.g., novel protein source or protein hydrolysate used in hypoallergenic diets), antibiotic medication (e.g., metronidazole), or sulfasalazine/mesalamine (5-aminosalicylic acid—5-ASA) for canine colitis [2,6,15,39]. Only a tiny percentage that do not respond to these therapies will require immunosuppressive treatment—dogs and cats with IBD [2]. Various immunosuppressive medications are used alone or in combination to induce and then sustain remission [40]. Glucocorticoids (prednisolone 1–2 mg/kg PO q12–24h or budesonide 1 mg/m^2^ PO q24h) and/or additional immunosuppressive drugs (cyclosporine or chlorambucil) are commonly used to treat dogs and cats with IBD, and azathioprine is also used in dogs [4,13,32,41]. In dogs with PLE, if there is no strong indication of an immune-mediated etiology, immunosuppressive therapy should not be a first option and should be carefully applied after a non-responsive nutrition modification trial [4,41,42]. The use of supplemental medication such as pro-/prebiotics, gastroprotectors, and antiemetics may also benefit IBD patients [43,44,45]. It is important to correct any possible imbalance, for example, Vitamin B12 deficiency and supportive care. For dogs with PLE that use prolonged corticosteroids, treatment with antiplatelet agents (e.g., clopidogrel) is indicated to prevent the thrombotic risk [4,29].

In human mild to moderately active IBD, oral 5-ASA (2–3 g/day) is the standard treatment, especially for colitis signs [46]. 5-ASA is an anti-inflammatory drug derived by sulfasalazine without sulfapyridine. In human patients with moderate to severe IBD or that did not respond to 5-ASA therapy in 2–4 weeks, corticosteroids are indicated for at least 6 to 8 weeks [47]. Other immunosuppressive drugs such as azathioprine, methotrexate and cyclosporine can be used as an alternative treatment option for human IBD, similarly to companion animals [2,48]. Antibiotherapy with metronidazole or ciprofloxacin is mostly used in cases of enterocutaneous and perianal fistulae [40,46,47]. Finally, biological therapy including monoclonal antibodies targeting pro-inflammatory cytokine inhibitors (TNFα and IL-12/23), α4β7 anti-integrins, and more recently Janus Kinase (JAK) inhibitors, also are fundamental therapies for humans as these cytokines play an important role in the pathogenesis of IBD [47,49]. Monoclonal antibodies would be also a potential treatment for canine and feline IBD but are currently not available in veterinary medicine, and the knowledge of the specific pathways addressed in companion animals IBD should be firstly well investigated [4,6]. Table 1 summarizes the treatment steps to follow for humans, dogs, and cats with IBD.

A small percentage of dogs with IRE do not respond well to medical therapy and are classified as having non-responsive enteropathy (NRE), which has a poor long-term prognosis and a high rate of euthanasia [50]. Alternative immunomodulatory treatments, such as fecal microbiota transplantation (FMT) or stem cell therapy can be promising therapies and they need further studies [51,52]. In humans, there is a stem cell therapy already licensed in the EU for the treatment of complicated perianal fistulas in adult patients with non-active or mildly active luminal Crohn’s disease—Darvadstrocel (Alofsel®), comprising a suspension of expanded human allogeneic ASCs [53].

## 3. Mesenchymal Stromal Cells (MSCs)

### 3.1. MSC Characterization

Stem cells can be classified based on their ability to differentiate and their source throughout the body in embryonic stem cells, adult stem cells, and also in induced pluripotent stem cells [5,54,55,56,57]. Embryonic stem cells are totipotent cells; they can generate cells from all three germinal layers (endoderm, mesoderm, and ectoderm) as well as extraembryonic annexes (placenta and umbilical cord) [54,55,58]. Adult stem cells are multipotent cells; they are only partly specialized and capable of producing a limited number of cell types from their own germinative layer [5,54,55]. Finally, induced pluripotent stem cells are created by reprogramming adult cells into pluripotent cells in the laboratory [59,60,61].

Mesenchymal stem cells (MSCs), also known as mesenchymal stromal cells, are undifferentiated, non-hematopoietic cells with the ability to self-renew and found in a variety of adult or extraembryonic tissues [62,63,64]. Due to their multipotency, they have the ability to differentiate under specific circumstances into a variety of cell types, including myocytes, osteoblasts, chondroblasts, tenocytes, cardiomyocytes, hepatocytes, neuronal cells, endothelial cells, and photoreceptors [5,64]. MSCs play a critical function in the regulation of the immune system despite not meeting the prearranged requirements as immune cells [5,65]. These cells are reported to be easy to isolate and feasible growth in in vitro culture and are able to secrete a wide range of pro- and anti-inflammatory factors, including prostaglandins, cytokines, and chemokines, which impact immune cells’ activity. They have exceptional pleiotropic features, including the capacity to move toward injury sites via chemotaxis, angiogenesis, growth factor synthesis, and antifibrosis [57,62].

MCSs can be isolated from a wide variety of tissues, but are most commonly collected from adipose tissue (AT) and bone marrow (BM), since they are relatively simple sources for obtaining cells [61,62,66,67]. For the collection of BM samples, surgery, general anesthesia, and a proper aseptic procedure are required, which is more invasive than obtaining AT samples [61,68]. According to some researchers, MSCs obtained from AT have a stronger proliferative capacity than MSCs obtained from BM. However, there is still no agreement in the literature, and further differences between the cell’s sources are detailed based on the donor’s age or passage number of cells [61,69].

In culture, MSCs have a primarily fusiform shape and are capable of adhering to plastic [70]. One of the MSCs’ identifying characteristics is the expression of cell surface markers. The International Society for Cellular Therapy developed the minimum standards for classifying human MSCs in order to reach a wider consensus on their universal description. The capacity to stick to plastic under controlled culture conditions, cell surface expression of CD73, CD90, and CD105 (>95%), and the lack of the hematopoietic stem cell markers CD45, CD34, CD14 or CD11b, and CD79 are some of these requirements [71,72,73]. Because of interspecies variations, it is known that human MSC markers could not be entirely compatible with those needed for canine and feline MSC identification [74]. Some studies have shown that most canine and feline MSCs express cell surface markers CD44 and CD90 and lack the expression of hematopoietic marker CD34 [75,76].

### 3.2. MSCs’ Mechanism of Action

Although the mechanism of action of therapeutic MSCs is still not fully known, it has attracted attention recently due to its growing use in immunological and regenerative medicine clinical trials on both human and animal patients [61,77,78]. The mechanism of action most likely differs between species and are affected by a variety of factors such as culture conditions or an inflammatory environment [33]. Adaptive immune cells (T and B lymphocytes) and innate immune cells (macrophages, dendritic cells, neutrophils, eosinophils, basophils, natural killer cells, and natural killer T cells) phenotype and function are both influenced by MSCs [77,79,80,81]. MSCs’ anti-inflammatory, immunomodulatory, and immunosuppressive capabilities serve as the foundation for their therapeutic use. These cells can evolve into the desired cell type, allowing the wounded area to be healed [61,62]. 

Despite the fact that MSCs do not have intrinsic immunosuppressive capacity, this ability is developed when cells are triggered by a proinflammatory environment, with the production of cytokines such as INFγ, TNF-α, and interleukin-1 β [26,82]. Additionally, they possess a remarkable capacity for immunomodulation through paracrine signaling, whereby they secrete a variety of molecules to nearby cells (cytokines, growth factors, and microvesicles that can carry a cargo of proteins and other bioactive molecules), or through cell-to-cell contact, which results in vascularization, cellular proliferation in damaged tissues, and a decrease in inflammation [5,83].

MSCs influence the innate immune system via three key mechanisms. Activating proinflammatory monocytes and macrophages is one of these methods. Macrophages can be change from M1 classic macrophages (with proinflammatory functions) in M2 anti-inflammatory macrophages in the presence of MSCs and their soluble factors such as IL-6, PGE_2_, TGF-β, and HGF [1,5,84,85]. MSCs can also block monocyte development into mature DCs by producing soluble factors such IL-6, PGE2, TGF-β, and HGF [83,85]. Finally, MSCs limit NK cell proliferation and cause polymorphonuclear cell and cytotoxic T cell apoptosis, which can be mediated by cell-to-cell contact or by soluble molecules such as PGE2 and IDO, but also TGF-β [5,26,84,86,87].

According to a number of studies, MSC infusion also influence the adaptive immune system by raising the frequency of regulatory T cells while decreasing the number of T cells that secrete inflammatory cytokines [63]. The effect of MSCs on T cells appears to be proportional to the MSC/T cell ratio: a high MSC/T cell ratio inhibits T cell proliferation, whilst a low MSC/T cell ratio may promote T cell growth [33], which is an essential issue in IBD treatment. They also suppress B cell growth through cell-to-cell interaction and through a cell cycle arrest in the G0/G1 phase [61,84].

The therapy can be autologous or allogeneic, depending on whether the stem cells are derived from the patient or from another individual of the same species [61]. Recent studies using xenogeneic stem cells from other species have yielded positive and safe outcomes.

## 4. Clinical Trials with MSCs

The immunomodulatory impact of MSC treatment has been documented in animal models and in humans during the last few years, with a considerable improvement in clinical presentation. Table 2 and Table 3 summarize the clinical trials conducted to date on dogs and cats with MSC therapy for IBD.

MSCs as an alternative therapy for IBD are still a relatively new concept. However, in clinical studies on humans with inflammatory gastrointestinal and immunological disorders, this medication has proved to be efficacious and safe. Most studies in human IBD use BM-MSCs, and ASCs have been increasingly employed for the local treatment of perianal fistulas that result as a complication of CD [93,94,95,96]. Table 4 summarizes published studies of intravenous MSC therapies in human IBD (CD and UC). 

## 5. Discussion

CIE in dogs and cats is classified into many entities based on their response to medical therapy. In contrast, IBD in both animals and human patients is a condition that may requires immunosuppressive medications, biologic drugs and, in some cases, surgery [2]. 

Over the last decade, tremendous research efforts have resulted in a greater knowledge of the multifactorial disease complex of canine and feline IBD compared to humans. Despite the differences in classification of the disease, all species’ etiopathogenesis culminate in a combination of genetic causes, environmental factors, microbiota imbalance, and mucosa immune defects. It is well recognized that the pathogenesis in humans and animals evolves an exacerbated reaction of lymphocytes to normal antigens, although the profile of released cytokines varies throughout species and even within investigations. 

The majority of human research has used BM-MSCs, and in dogs and cats, ASCs. Adipose tissue is presently the most accessible and abundant source of MSCs, with a less painful collecting technique when compared to other sources. In Veterinary Medicine, adipose tissue is the most frequent source for stem cells obtainment when the MSC option is considered for IBD treatment. It is often obtained from animals undergoing routine surgery (e.g., ovariohysterectomies), where small quantities of adipose tissue are regularly discharged. Additionally, BM-MSCs are affected by age, and increasing age decreases cell number, proliferation, and differentiation capacities [109]. ASCs, on the other hand, show more growth potential without losing their differentiation potential, resulting in massive cell numbers that are increasingly being used in cell therapy [110]. 

Studies carried out with MSCs in dogs and cats used doses between 2 and 4 million cells per kg/animal. Intravenous allogeneic cell administration was used in all studies, and in certain trials, MSC therapy was compared to conventional prednisolone therapy. ACSs safely increased life quality of dogs receiving or not receiving prednisolone concomitantly to stem cell therapy. In cats, Webb and Webb (2022) showed that therapy alone with prednisolone or therapy with MSCs appears to be equal effective for feline IBD therapy [91]. Cats improved in clinical symptoms and scores when given ASCs or prednisolone, although the adverse effects of prolonged corticosteroid use are clearly severe. In human studies, dosages ranging from 1 to 3.5 million cells per kg/person were employed, and results have shown a safe and feasible therapy, with minimal or no side effects, possible with no association to stem cell administration. One study also demonstrated safety and efficacy in treatment with MSCs at doses of up to 10 million cells per kg/person [103].

## 6. Conclusions and Future Directions

Mesenchymal stromal cells appear to be a successful and safe therapy for IBD in humans, dogs, and cats. Future research is needed to draw more conclusions about these cell therapies, using more standardized techniques and larger patient cohorts. Other regenerative therapies have also been recently studied for IBD patients, such as stem cells’ secretome, stem cells’ extracellular vesicles, and organoids. MSCs’ therapeutic effects occur through the production and release of trophic molecules, which is now referred to as the secretome [111]. The MSC secretome is a complex combination of soluble products comprising a proteic soluble fraction (growth factors and cytokines) and a vesicular fraction (microvesicles and exosomes). Therapies without cell transplantation, such as the one offered by MSC’s secretome, can offer key advantages over transplanting stem cells, although they are still poorly studied [111].

Exosomes are one type of extracellular vesicle that is now being researched as a potential therapy for IBD patients [112]. Bilayer vesicular nanoparticles known as exosomes are released by cells as a result of environmental stimulation or self-activation and exhibit the same features as their parental cells. Exosomes produced from IECs have the ability to trigger immunoregulatory pathways to promote the maturation of macrophages with tolerogenic features, regulatory T cells (Treg), and DCs, as well as to maintain intestinal homeostasis [112]. Extracellular vesicles produced from adipose tissue-derived MSCs have been shown to be effective against dextran sulfate sodium-induced colitis in mice. Nevertheless, the underlying processes have yet to be fully clarified [113].

Finally, organoids are sophisticated three-dimensional cell culture models created from stem cells that have a significantly greater ability for self-organization and reproduction of in vivo physiology than two-dimensional cell culture systems [114,115]. Even though organoid systems may accurately mimic host biology, they may be restricted in research evaluating nutrients and drugs transport or host–microbiome interactions [116]. Intestinal organoids have been used as a model to study the pathogenesis of IBD as they closely mimic the structure and behavior of the original organ. Ultimately, these organoids can be transplanted into IBD patients to repair epithelial damage and reverse all symptoms of the disease. This is expected to occur in the very near future.

## Figures and Tables

**Figure 1 biomedicines-12-00735-f001:**
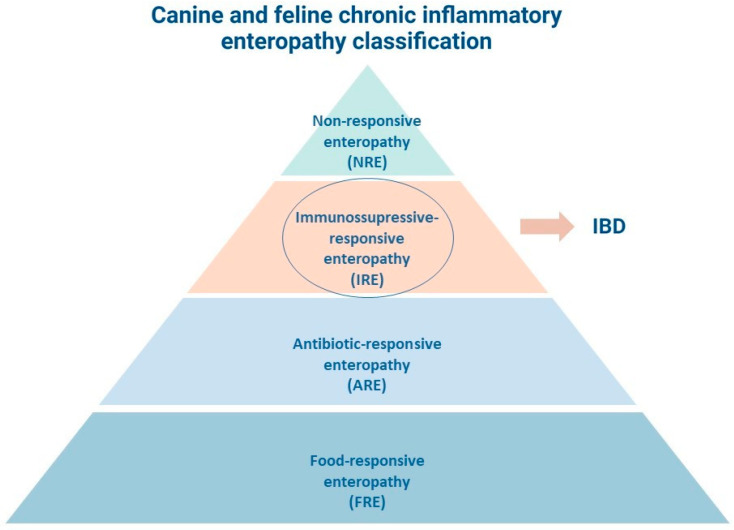
Chronic inflammatory enteropathy classification in dogs and cats according to treatment response.

**Figure 2 biomedicines-12-00735-f002:**
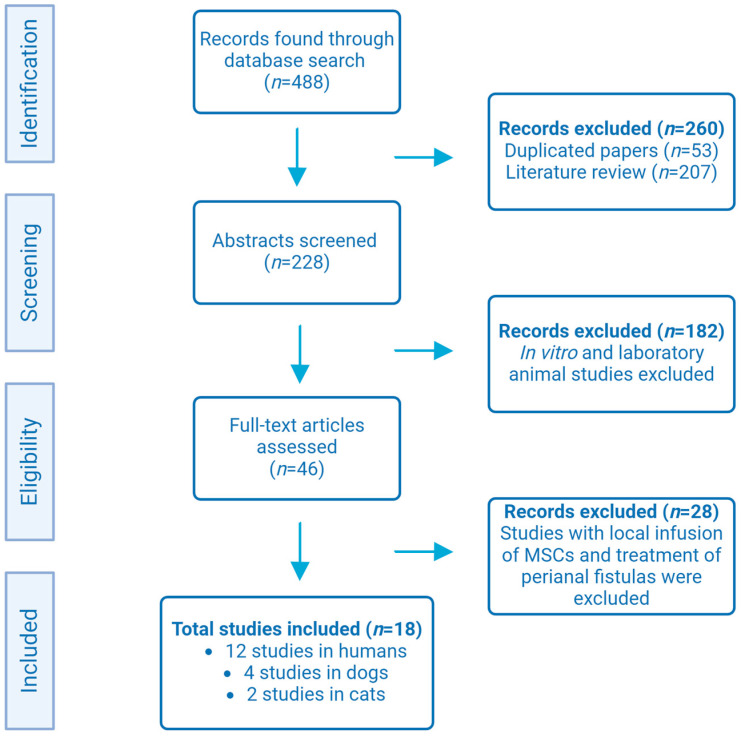
Flowchart of study selection. A total of 488 studies were analyzed. There were 18 studies that met the following criteria: clinical trials in humans, dogs, and cats with IBD treated with IV infusion of MSCs (Original figure created with BioRender.com).

**Figure 3 biomedicines-12-00735-f003:**
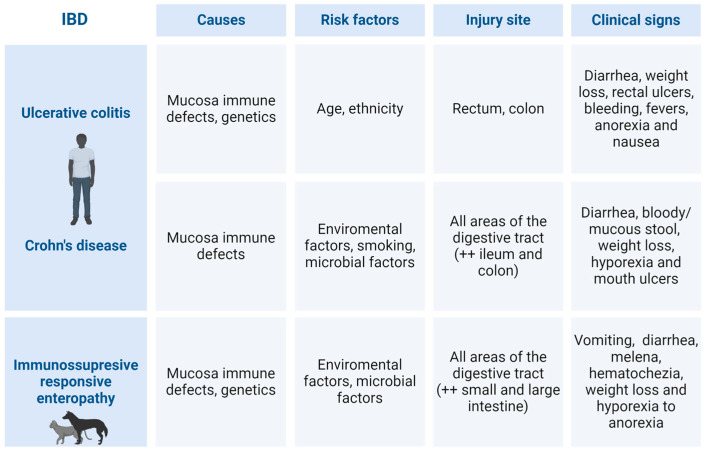
Differences in IBD causes, risk factors, injury sites and clinical signs between human and companion animals (dogs and cats). ++—with special incidence. (Original figure created with BioRender.com).

**Figure 4 biomedicines-12-00735-f004:**
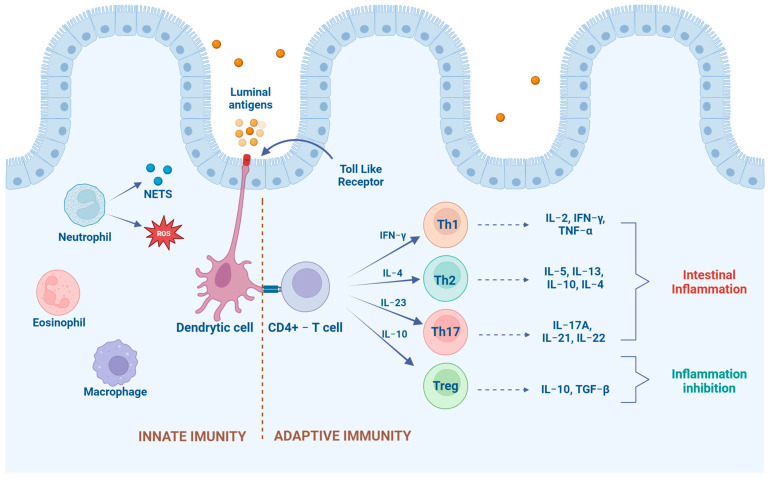
Immune response mechanisms in the pathogenesis of IBD (Original figure created with BioRender.com).

**Table 1 biomedicines-12-00735-t001:** IBD treatment in humans, dogs, and cats.

HUMANS	DOGS/CATS
**1. 5-ASA**	**1. Diet modification** (hypoallergenic)
**2. Antibiotics** (Metronidazole), **Corticosteroids** (Prednisolone, Budenoside)	**2. Antibiotics** (Metronidazole, Tylosine, Oxytetracycline)
**3. Immunosuppressants** (Azathioprine, Methotrexate, Cyclosporine)	**3. Corticosteroids** (Prednisolone, Budenoside)
**4. Biological therapy** (TNF-α and IL-12/23 blockers, α4β7 anti-integrins and JAK inhibitors	**4. Immunosuppressants** (Cyclosporine, Clorambucil, Azathioprine)

**Table 2 biomedicines-12-00735-t002:** Clinical trials carried out with MCSs on canine IBD.

Nº Dogs	Treatment Group	Control Group	Outcomes	Ref.
N = 11 dogs	Single IV allogeneic ASCs (2 × 10^6^ cells/kg)	N/A	After 2 weeks of treatment, a clinical response occurred in all dogs. ASCs were well tolerated and seemed to have therapeutic advantages.	Pérez-Merino et al., 2015 [88]
N = 32 dogs	19 dogs received a single IV allogeneic ASCs infusion (4 × 10^6^ cells/kg)	13 dogs received the same ASCs treatment with prednisolone simultaneously	Clinical improvement was somewhat lower in the group receiving combination treatment. All animals improved in one or more signs. ACSs safely increased life quality of dogs receiving or not receiving prednisolone.	Cristóbal et al., 2021 [11]
N = 16	Single IV allogeneic ASCs (4 × 10^6^ cells/kg)	N/A	Neutrophil-to-lymphocyte ratio (NLR), platelet-to-lymphocyte ratio (PLR), and systemic immune-inflammation index (SII) decreased significantly after 2 months of therapy, while NLR and SII reached normal levels at 9 months. Changes in blood inflammatory markers were associated with significant clinical improvement.	Cristóbal et al., 2022 [89]
N = 32	Nine dogs received a single IV allogeneic ASCs infusion (4 × 10^6^ cells/kg) and 11 dogs received the same amount of ASCs with prednisone	12 healthy dogs	Treatment with MSCs, either alone or in combination with corticosteroids, was shown to be effective. The oxidative markers albumine, malondialdehyde, and glutathione were measured before and after therapy. Only the albumin value changed (increased) following the MSC therapy.	Cristóbal et al., 2023 [90]

IV—intravenous; ASCs—adipose-derived mesenchymal stromal cells; NLR—neutrophil-to-lymphocyte ratio; PLR—platelet-to-lymphocyte ratio; SII—systemic immune-inflammation index.

**Table 3 biomedicines-12-00735-t003:** Clinical trials carried out with MCSs on feline IBD.

Nº Cats	Treatment Group	Control Group	Outcomes	Ref.
N = 7	Two IV allogeneic ASCs (2 × 10^6^ cells/kg)	N/A	Clinical signs improved in five of seven cats treated with MSCs. ASCs was well tolerated and appeared to produce clinical advantages.	Webb and Webb 2015 [91]
N = 12	Six cats received two IV allogeneic ASCs (2 × 10^6^ cells/kg)	Six cats received treatment with prednisolone	Cats that completed the research showed improvements in clinical signs and scores, both with ASCs and prednisolone. Treatment with ACSs appears to be as effective in the treatment of feline IBD as prednisolone therapy.	Webb and Webb 2022 [92]

IV—intravenous; ASCs—adipose-derived mesenchymal stromal cells; N/A—not applicable; IBD—inflammatory bowel disease.

**Table 4 biomedicines-12-00735-t004:** Clinical trials carried out with MCSs on human IBD.

Nº Patients	Disease	Treatment Group	Control Group	Outcomes	Ref.
10	CD	Two IV autologous BM-MSC infusions (1–2 × 10^6^ cells/kg) with 7 days apart	N/A	Clinical response was achieved in 3 of 10 patients (CDAI > 70) 6 weeks post treatment, but none achieved remission (CDAI < 150).	Duijvestein et al., 2010 [97]
50	CD and UC	Single IV allogeneic BM-MSC infusion (1.5–2.0 × 10^8^ cells/patient) (39 UC and 11 CD)	Traditional treatment (30 UC and 10 CD patients).	The MSC-treated group had a significant reduction in indices of clinical and morphological activity of an inflammatory process compared to traditional treatment group, and clinical remission occurred in 40 patients (80%).	Lazebnik et al., 2010 [98]
9	CD	Autologous peripheral blood stem cell transplantation (PBSCT)	N/A	Five patients experienced a clinical and endoscopic remission within 6 months after treatment. Relapses occurred in 7/9 patients over the follow-up period, although they were managed with conventional therapy.	Hasselblatt et al., 2012 [99]
7	CD And UC	Single IV allogeneic BM-MSC/UC-MSC infusion (1 × 10^6^ cells/kg)	N/A	All patients had a significant reduction in CDAI. Five patients achieved remission at 3 months post treatment.	Liang et al., 2012 [100]
50	CD	30 treated with allogeneic IV BM-MSC infusions (3 × 10^6^ cells/kg two times over 1 month with 1-week interval and then every 6 months)	20 patients receiving an anti-TNF agent (infliximab)	In the MSC group, complete clinical remission was achieved in 21 patients after 2 months, and clinical and endoscopic remission was achieved in 17 patients after 6 months. In 4 years, CDAI in the infliximab group was significantly lower than in the MSC group.	Knyazev et al., 2013 [101]
15	CD	Four IV allogeneic BM-MSC infusions (2 × 10^6^ cells/kg weekly for 4 weeks)	N/A	CDAI decreased by > 100 in 12 of 15 patients (80%) at 42 days post treatment and clinical remission (CDAI < 150) occurred in 8 of 15 patients.	Forbes et al., 2014 [102]
12	CD	Single IV autologous BM-MSC infusion (three different doses: 2, 5, and 10 × 10^6^ cells/kg)	N/A	5/11 experienced clinical response (decrease in CDAI by ≥ 100 points at 2 weeks). MSC infusions of up to 10 million cells/kg have been shown to be safe and well tolerated.	Dhere et al., 2016 [103]
70	UC	34 patients received two IV UC-MSC infusions (first: 0.5 × 10^6^ cells/kg and second: 1.5 × 10^6^ cells/kg	36 patients received a saline solution	At 3 months, 29/34 (85.3%) patients in the MSC group showed clinical response (a drop in total Mayo UC activity score of ≥3 and ≥30% from baseline) compared to 6/36 (16.7%) in control group.	Hu et al., 2016 [104]
22	UC	12 patients received Autologous IV BM-MSC infusions (1.5–2 × 10^6^ cells/kg at weeks 0, 1, and 26)	10 patients received standard anti-inflammatory therapy	After 3 years, the treatment group had a 50% remission rate (6/12) compared to the control group’s 10% (1/10).	Knyazev et al., 2016 [105]
13	CD	Two autologous IV BM-MSC infusions (3.55 ± 0.45 × 10^6^ cells/kg at weeks 0 and 4)	N/A	At 8 weeks, 2/13 (15.4%) experienced clinical response, with a CDAI drop of ≥100 points. There was no infusion toxicity or significant adverse event identified.	Gregoire et al., 2018 [106]
34	CD	Group 1 (15 patients) received Allogeneic IV BM-MSC infusion (2 × 10^6^ cells/kg with AZA 2–2.5 mg/kg at months 0, 1, and 6)	Group 2 (19 patients) received the same MSC treatment without AZA	Clinical remission (CDAI < 150) at 12 months in both groups. The combination with AZA produces a stronger anti-inflammatory effect.	Knyazev et al., 2018 [107]
82	CD	41 patients received IV UC-MSC infusions (1 × 10^6^ cells/kg, with one infusion per week for 4 weeks)	41 patients received standard anti-inflammatory therapy	12 months following therapy, the CDAI, HBI, and corticosteroid dose had reduced.	Zhang et al., 2018 [108]

CD—Crohn’s disease; IV—intravenous; UC—ulcerative colitis; BM-MSCs—bone marrow-derived mesenchymal stem cells; N/A—not applicable; CDAI—Crohn’s disease activity index; MSC—mesenchymal stem cell; PBSCT—peripheral blood stem cell transplantation; UC-MSCs—umbilical cord-derived mesenchymal stem cells; TNF—tumor necrosis factor, AZA–azathioprine.

## Data Availability

Not applicable.

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
