# Peer review of "A Comprehensive Exploration of Therapeutic Strategies in Inflammatory Bowel Diseases: Insights from Human and Animal Studies"

_biomedicines, 2024, doi:10.3390/biomedicines12040735_

Round 1

Reviewer 1 Report

Comments and Suggestions for Authors

This comparative review of data from dogs, cats and humans with IBD is an interesting update on the presentation and treatment of the disease in different species. Some suggestions to improve the text are presented below:

Line 74: Immunosuppressive of immunomodulatory therapy is not always required for IBD patients. In fact IBD, particularly ulcerative colitis, can be treated long term with anti-inflammatory drugs consisting in mesalacine and other drugs derived from 5-aminosalicilates. Corticosteroids are also used to induce remission in both Crohn’s disease and ulcerative colitis. Dietary therapy, namely exclusive enteral nutrition, has demonstrated effectiveness to induce remission in pediatric IBD. This sentence should be modified to include these considerations.

Lines 190-191: Cytokine inhibitors, such as TNFα and IL-12/23, are not a promising therapy but an standard for care for humans with IBD, especially those who suffer from Crohns’ as around 1/3 of patients receive this biological therapy, alone or combined with other immunomodulatory drugs. Please consider to re-write your sentence.

Table 1. Please consider adding to human therapies Janus kinase inhibitors (JAK inhibitors) as some of them are approved and used to treat IBD; among biological drugs, add also α4β7 anti-integrins and anti-IL23 drugs, as they are also used in these patients.  

Regarding to allogenic adipose tissue-derived stem cells, there is a therapy already approved, marketed and used in patients with Crohn’s disease, called darvadstrocel, which is used in patients with complicated, refractory fistulas, applied inside the fistulous tract. Information on this advance should be added to a revised version of the manuscript.

Lines 341-342: I suggest: “In contrast, IBD in both animals and human patients is a condition that may requires immunosuppressive medications, biologic drugs and, in some cases, surgery” as many patients with ulcerative colitis will maintain disease in remission long term with anti-inflammatory 5-aminosalycilates.

Author Response

Response to reviewer #1.

Dear reviewer, Dear colleague, we thank you for your careful reading and the important suggestions for improving the manuscript.

This comparative review of data from dogs, cats and humans with IBD is an interesting update on the presentation and treatment of the disease in different species. 

Some suggestions to improve the text are presented below:

Line 74: Immunosuppressive of immunomodulatory therapy is not always required for IBD patients. In fact IBD, particularly ulcerative colitis, can be treated long term with anti-inflammatory drugs consisting in mesalacine and other drugs derived from 5-aminosalicilates. Corticosteroids are also used to induce remission in both Crohn’s disease and ulcerative colitis. Dietary therapy, namely exclusive enteral nutrition, has demonstrated effectiveness to induce remission in pediatric IBD. This sentence should be modified to include these considerations.

Done between lines 74 and 76. Other ideas already appear in the text elsewhere, such as "Corticosteroids are also used to induce remission in both Crohn’s disease and ulcerative colitis. " in lines 195 to 197.

Lines 190-191: Cytokine inhibitors, such as TNFα and IL-12/23, are not a promising therapy but an standard for care for humans with IBD, especially those who suffer from Crohns’ as around 1/3 of patients receive this biological therapy, alone or combined with other immunomodulatory drugs. Please consider to re-write your sentence.

We are grateful for the reviewer's recommendation, which now appears in the text between lines 200 and 203.

Table 1. Please consider adding to human therapies Janus kinase inhibitors (JAK inhibitors) as some of them are approved and used to treat IBD; among biological drugs, add also α4β7 anti-integrins and anti-IL23 drugs, as they are also used in these patients.  

We are grateful for the reviewer's recommendation, which now appears in the Table 1 near line 211.

Regarding to allogenic adipose tissue-derived stem cells, there is a therapy already approved, marketed and used in patients with Crohn’s disease, called darvadstrocel, which is used in patients with complicated, refractory fistulas, applied inside the fistulous tract. Information on this advance should be added to a revised version of the manuscript.

We are grateful for the reviewer's recommendation, which now appears in the text between lines 216 and 219.

Lines 341-342: I suggest: “In contrast, IBD in both animals and human patients is a condition that may requires immunosuppressive medications, biologic drugs and, in some cases, surgery” as many patients with ulcerative colitis will maintain disease in remission long term with anti-inflammatory 5-aminosalycilates.

Done at Lines 365 to 366

Best regards

The authors

Reviewer 2 Report

Comments and Suggestions for Authors

Mesenchymal stem cell (MSC) therapy is emerging as a promising and innovative avenue for Inflammatory Bowel Diseases (IBDs) treatment. As the authors assert, the pathogenesis of IBD is intricate and remains under active investigation. The current review provides a detailed examination of the existing treatments for IBD in both animal and human subjects.

The primary objective of this review is to compare the efficacy of mesenchymal stem cell treatment for IBD in animal and human studies. However, it is noteworthy that the section dedicated to human studies lacks a comprehensive review of ongoing trials. To enhance the depth and breadth of the review, it is advisable for the authors to conduct a more exhaustive examination of human trials involving MSC treatment for IBD. This additional analysis will contribute significantly to our understanding of the therapeutic potential in human subjects.

Furthermore, the authors are encouraged to improve the coverage of animal studies by addressing the completeness of the trials. This improvement could involve a more detailed analysis of the methodologies, outcomes, and limitations of the animal studies, thereby enhancing the overall robustness of the review.

Suggested Revisions:

Title may revised, such as “A Comprehensive Exploration of Therapeutic Strategies in Inflammatory Bowel Diseases: Insights from Human and Animal Studies"

Add a dedicated section to ongoing human trials of MSCs in IBD, complemented by a table or forest plot summarizing key findings. This addition will significantly augment the review's relevance and utility to readers interested in the current state of clinical trials in this field.

Comments on the Quality of English Language

moderate revise  or edit after revise

Author Response

Dear reviewer

We are very grateful for the relevant contribution and encouragement that led us to change title and introduce several tables into the manuscript that illustrate the most relevant clinical trials relating to the topic under analysis.

Best regards

Round 2

Reviewer 1 Report

Comments and Suggestions for Authors

Thank you for your review of the manuscript. I think its content is now more precise